

**Investigating recent decadal trends in the Pacific westerly jet in response to various**
**atmospheric forcings using CMIP6 model results and reanalysis data**
Huisheng Bian[1,2], Sarah Strode[3,2], Mian Chin[2], Fan Li[1,2], Andrea Molod[2], Peter R Colarco[2], and
Hongbin Yu[2]
[1]Goddard Earth Sciences Technology and Research (GESTAR) II, UMBC, Baltimore, MD,
21250, USA
[2]Laboratory for Atmosphere, NASA Goddard Space Flight Center, Greenbelt, MD, 20771, USA
[3]Goddard Earth Sciences Technology and Research (GESTAR) II, MSU, Baltimore, MD, 21251,
USA
*Correspondence to*: Huisheng Bian (Huisheng.Bian@nasa.gov) and Mian Chin
(mian.chin@outlook.com)
**Abstract**
The strength and location of the North Pacific westerly jet (NPWJ) strongly affects weather and
trans-Pacific pollution transport as it triggers and directs continuous atmospheric river events
toward North America. In this study, we used four reanalysis datasets and eight Coupled Model
Inter-comparison Project Phase 6 (CMIP6) models to investigate the characteristics and changes
of the NPWJ during 1980-2019. The NPWJ climatologic core seasonally swings between north
and south, being most southward (~33°N) in winter and most northward (~45°N) in summer,
as shown by the observation-based reanalysis data. All reanalysis and CMIP6 data provide strong
evidence for the weakening (up to -0.45 and -0.68 m s$^{-1}$ decade$^{-1}$) and northward shift (0.2° and
1.0°) of the NPWJ in summer and autumn during the study period. Various atmospheric forcing
experiments performed by the CMIP6 models further reveal aerosol forcing being the main
driver, which can be traced back to the spatially inhomogeneous anthropogenic aerosol emission
changes that increase in Asia and decrease in Europe. When we apply Earth system climate
models to investigate the feedback between atmospheric forcings and atmospheric dynamical
fields on decadal scales, two points should be noted. First, there is a need to include interactive
chemistry in the CMIP6 model simulations to bring the dynamical fields closer to those based on
observational data. Second, in addition to the well-mixed greenhouse gases, anthropogenic
aerosols, and natural forcings proposed in the Detection and Attribution Model Intercomparison
Project (DAMIP) single-forcing simulations, time-varying ozone radiative forcing is also
important to climate change.

**1. Introduction**
The westerly jet stream is a year-round fast-flowing current of air, circling the Earth between the
Arctic and mid-latitudes. This system, along with the total kinetic energy of synoptic storm
systems and the number of strong cyclones, determines the mid-latitude synoptic circulation
(Coumou et al., 2015, 2017; Chang et al., 2016). Differences in regional mean warming status
are changing the behavior of the jet stream in a way that favors more extreme and persistent
weather anomalies. For the past four decades, a wavier jet circulation has been detected,
coinciding with accelerated Arctic warming and a reduced near-surface meridional temperature
gradient (MTG) (Blackport and Screen, 2020). When the jet stream takes larger north-south
meanders – known as "Rossby waves" – warm air can penetrate into the Arctic, and cold air can





plunge southward. Larger amplitudes of the waves also mean the systems moving from west to
east tend to travel more slowly relative to smaller amplitudes, effectively making weather
conditions lingering and more persistent and inter-continental pollution transport more sluggish.
Studies of such jet variation and impact have been performed, focusing primarily on the Atlantic
during winter and summer seasons (Bracegirdle et al., 2021, Hall et al., 2016, Iqbal et al., 2018,
Kwon et al., 2018, Linderholm et al., 2017, Osman et al., 2021, Rousi et al., 2021, Trouet et al.,
2018, Viillings et al., 2012, 2013).  A previous study using the Coupled Model Intercomparison
Project Phase 6 (CMIP6) simulations provided compelling evidence that changes in
anthropogenic aerosol precursor emissions were the primary driver of the weakening of the
summer Eurasian subtropical westerly jet (ESWJ) over the last four decades (Dong et al., 2022).
But northern hemisphere jet variability and trends differ on a regional basis (e.g., North Atlantic,
North Pacific, etc) and on seasonal to decadal timescales, suggesting that different mechanisms
are influencing jet position and speed (Mann et al., 2017). Compared to the North Atlantic, the
North Pacific is observed to have larger interannual jet variability (Hallam et al., 2022),
suggesting a necessity of looking into the characteristics of the North Pacific Westerly Jet
(NPWJ) separate from the Atlantic Westerly Jet. Different regions of the world are seeing
different levels of the effects associated with regional trends of anthropogenic aerosol emissions.
Unlike the decreasing trends over North America and Europe, anthropogenic emissions over
Asia increased significantly during 1980-2010 resulting from rapid economic growth and
decreased afterward owing to strict emission control primarily in China. Although assessing the
effects of anthropogenic aerosols on the Pacific storm track via aerosol-cloud-radiation
interaction have been performed using aerosols in winter season (Zhang et al., 2007) and in years
of 2000 and 1850 (Wang et al., 2014), the interactions between human induced aerosol and
westerly jet variation in that region on a multi-decadal scale remain undetermined.
The profound impacts of the NPWJ changes on weather, air pollution, and climate make the
study of NPWJ necessary. It is an important atmospheric feature that initiates and directs
sequential atmospheric river events toward California and beyond. Its strength and location
regulate extreme weather events, such as a reduced number of strong extratropical cyclones
(Chang et al., 2016; Zhao et al., 2020), high precipitation (Fish et al., 2022; Wang et al., 2017),
severe drought (Wang et al., 2017; Swain et al., 2016; 2017), high-fire (Guirguis et al., 2022,
Monroe 2022; Wahl et al., 2019), and prolonged summer heat extremes across California (Swain
et al., 2016), water availability and flood risk in western U.S. (Gonzales et al., 2020), cold air in
the midwest and the central/eastern U.S. (Flis 2022), and tornadoes and other severe weather in
the southern U.S. (Flis 2022).  It also regulates the trans-Pacific transport of atmospheric
pollutants.
In this study, we investigate the recent decadal trends of the NPWJ attributable to various
atmospheric forcings through integrating observations and models. We will address the
following science questions: 1. What is the strength and location of the westerly jet over the
North Pacific during 1980-2019? 2. What are the trends in the jet strength and location, and how
do they vary seasonally? 3. How does the NPWJ respond to various atmospheric forcings and
which atmospheric forcing is dominant (or most important)?





Section 2 describes four sets of reanalysis data and eight CMIP6 model results. We then use
these data to study the NPWJ in Section 3, focusing on the strength, location, and changes/trends
of the NPWJ using reanalysis data in Section 3.1 and studying the responses of NPWJ changes to
various atmospheric forcings using CMIP6 model results in Section 3.2. Section 4 discusses the
uncertainties in using the reanalyses and CMIP6 data in the study. Finally, Section 5 summarizes
the current research and proposes future studies.
**2.  Data Description**
Four reanalysis datasets that combine vast amounts of historical observations into global
estimates using advanced modeling and data assimilation systems are used to provide
observational constraints for the strength and location of the NPWJ and its trend during the
modern satellite era. These four datasets are (1) the European Centre for Medium-Range
Weather Forecast (ERA5) (Hersbach et al., 2020); (2) the Japanese 55-year Reanalysis Project
(JRA55) (Kobayashi et al., 2015); (3) the Modern-Era Retrospective Analysis for Research and
Applications, version 2 (MERRA2) (Gelaro et al., 2017); and (4) the National Center for
Environmental Prediction (NCEP Reanalysis) (Kalnay et al., 1996). These reanalysis data
provide observational constrains since they incorporate conventional ground and aircraft
observations, as well as extensive satellite retrievals including MODIS winds, MLS temperature
and ozone, and OMI total column ozone. The MODIS retrieved aerosol AOD is also assimilated
in MERRA2. Although these datasets start at different years, they all cover the period of 1980 to
present. Details on each dataset's spatial resolution, use of aerosol data assimilation, and data
locations are summarized in Table 1. We will use daily and monthly mean zonal winds to
describe the NPWJ.
Table 1. Information of the four-reanalysis data used in this study

| Reanalysis Product[1] | Starting Year | Assimilation AOD | Spatial Resolution (lon, lat, lev) | Data Access |
|---|---|---|---|---|
| ERA5 | 1940 | Yes | 0.25x0.25, 37 levels | https://climate.copernicus.eu/climate-reanalysis |
| JRA55 | 1958 | No | 0.563x0.562, 60 levels | https://jra.kishou.go.jp/JRA-55/index_en.html#reanalysis |
| MERRA2 | 1979 | Yes | 0.625x0.5, 72 levels | https://gmao.gsfc.nasa.gov/reanalysis/MERRA-2/data_access/ |
| NCEP | 1948 | No | 2.5x2.5, 17 levels | https://psl.noaa.gov/data/gridded/data.ncep.reanalysis.html |

[1]The four-reanalysis data on 1980-2019 are used in this study.
Results from eight Coupled Model Intercomparison Project Phase 6 (CMIP6) models will be
used to provide multi-model, multi-ensemble simulations for atmospheric dynamics in
atmosphere-ocean coupled Earth system models. The CMIP6 experiments used in this study are
the pre-industrial control simulation (piControl, aka piCtl hereafter) used to identify Earth
system internal variability (Collins et al., 2017), the CMIP historical all- forcing (ALL)
simulations (Eyring et al., 2016) which time-varying have forcings evolving from pre-industrial
conditions to 2014, and the Detection and Attribution Model Intercomparison Project (DAMIP)
single forcing simulations (Gillett et al., 2016). Single forcing experiments include GHG
(GreenHouse Gas) only (driven with changes in well-mixed greenhouse gas concentrations
only), AER (AERosol) only (driven with changes in anthropogenic aerosol emissions), and NAT
(NATural) only (driven with changes in natural forcings including solar irradiance, land use,





etc.) simulations which were designed to estimate the contributions of different anthropogenic
and natural forcings to observed global and regional climate changes. Since the CMIP6 model
simulations are long-term free-running (i.e., not constrained by observed meteorology) General
Circulation Model (GCM) simulations, we choose only CMIP6 models that have at least three
ensemble simulations for all historical and single forcing simulations to use in this study. Eight
CMIP6 models meet this requirement and an ensemble analysis of each of these models is
performed. They are the Beijing Climate Center Climate System Model (BCC-CSM2-MR) (Wu
et al., 2019; 2021), the Canadian Earth System Model version 5 (CanESM5) (Swart et al., 2019),
the sixth generation Centre National de Recherches Météorologique Coupled Model (CNRM-
CM6-1) (Voldoire et al., 2019), the Goddard Insitute for Space Studies climate model (GISS-E2-
1-G) (Kelley et al., 2020), the Hadley Centre Global Environment Model version 3 (HadGEM3-
GC31-LL) (Williams et al., 2018), the Institute Pierre-Simon Laplace Climate Model (IPSL-
CM6A-LR) (Boucher et al., 2020), the Model for Interdisciplinary Research on Climate version
6 (MIROC6) (Tatebe et al., 2019), and the Meteorological Research Institute Earth System
Model (MRI-ESM2-0) (Yukimoto et al., 2019). Information of the eight CMIP6 results in terms
of their spatial resolutions and ensemble numbers is summarized in Table 2. The CMIP6
monthly three-dimensional distribution of zonal wind (U) and temperature (T), downward
surface solar radiation (SSR), surface air temperature (SAT), and aerosol optical depth (AOD)
are used in the form of multi-model mean (MMM) data constructed on top of the ensemble data
from each model.
Table 2. Information of the eight CMIP6 models used in this study

| Designed Simulation[1] | | CMIP6 Model | Spatial Resolution (lon, lat, lev) | Ensemble Run | Note |
| Long Name | Short Name | | | | |
|---|---|---|---|---|---|
| CMIP/historical CMIP/piControl[2] | hist piCtr | BCC-CSM2-MR | 1.125x1.125x46 | r1,r2,r3 | r3: has only hist-GHG |
| | | CanESM5 | 2.8x2.8x49 | r1,r2,r4,r5,r6,r7,r8,r9,r10 | r3: not has hist-GHG |
| | | CNRM-CM6-1 | 1.4x1.4x91 | r1,r2,r4,r5,r6,r7,r8,r9,r10 | All r has not hist-nat |
| DAMIP/hist-GHG DAMIP/hist-aer DAMIP/hist-nat | GHG AER NAT | GISS-E2-1-G | 2.5x2.0x40 | r1,r2,r3,r4,r5 | Has more rxx |
| | | HadGEM3-GC31-LL | 1.875x1.241x85 | r1,r2,r3,r4,r5 | |
| | | IPSL-CM6A-LR | 2.5x1.26x79 | r1,r2,r3,r4,r5,r6,r7,r8,r9 | |
| | | MIROC6 | 1.4x1.4x81 | r1,r2,r3 | |
| | | MRI-ESM2-0 | 1.125x1.125x80 | r1,r3,r5 | r2: not has ua in hist-GHG |

[1]CMIP6 data on 1980-2014 are used in this study.
[2]piControl has only r1 simulation.
**3.    Results and discussions**
**3.1 Investigation of the strength, location, and trend of the Pacific westerly Jet using four**
**reanalysis datasets**
We first use the four reanalysis datasets to answer our first two science questions in Section 1
regarding the strength and location of the NPWJ over the period of 1980-2019 and their trends
and seasonal variability.
A statistical analysis for NPWJ strength, location, and trend are performed by introducing the Jet
Latitude Index (JLI) via a modification of the approach used in Woollings et al. (2010) and
Davini et al., (2014). Analyses are performed focusing on the North Pacific area (120°E - 240°E,
30°N – 45°N, afterward NPA) during the period 1980-2019 on daily and seasonal basis. The
algorithm calculates the latitude and speed of the jet stream as follows:



1. Daily mean zonal winds are zonally averaged along a longitude segment of the NPA with
vertical averages ranging from 300 hPa to 150 hPa, covering the vertical range of maximum
zonal winds centered around 200 hPa.
2. We keep the features associated with synoptic systems by applying a 5-day running-mean
filter to remove the features associated with individual small-scale perturbations.
3. The maximum westerly wind speed of the resulting profile is then identified and defined as the
jet speed. The JLI is defined as the latitude at which this maximum is found.

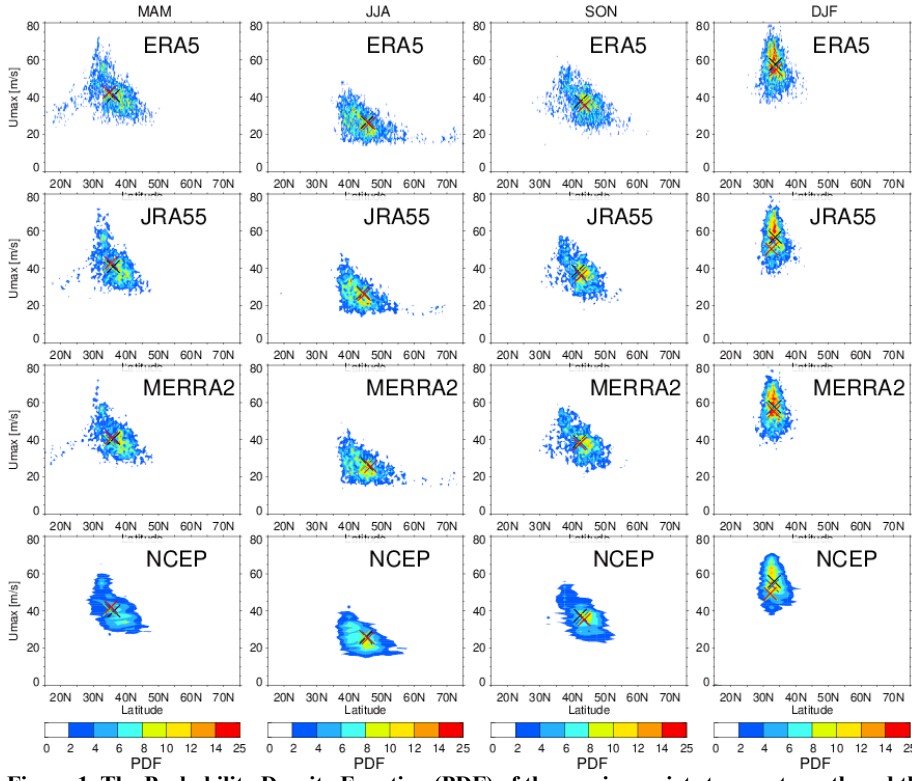

**Figure 1. The Probability Density Function (PDF) of the maximum jet stream strength and the corresponding jet stream latitude derived from the JLI analysis using daily zonal wind U composited on North Pacific (120°E - 240°E) centered around 200 hPa during 1980-2019 for four seasons (MAM, JJA, SON, and DJF) based on the four reanalysis datasets. The black and red cross symbols show the maximum U and its corresponding latitude in the first (1980s) and last (2010s) decades.**

The Probability Density Function (PDF) of the maximun jet stream strength and the
corresponding jet stream latitude derived from the JLI analysis for the four seasons from 1980 to
2019 using the four reanalysis datasets is shown in Fig. 1. The figure shows clearly the strength
and location of the jet stream, and the relationship between jet strength and location on seasonal
basis, i.e. March-April-May (MAM), June-July-August (JJA), September-October-November
(SON), and December-January-February (DJF). The strength, location, and seasonal variation of
the NPWJ from the four reanalysis data are remarkably similar. Basically, the NPWJ is strongest
in winter, gradually weakens toward summer, and then turns around to strengthen from summer
to winter. Meridionally, the NPWJ center is at the southernmost point in winter, moves





northward to the northernmost point in summer, and then returns to the southernmost point in
winter (i.e., about 33°N, 38°N, 45°N, and 43°N for DJF, MAM, JJA, and SON, respectively).
The maximum zonal wind speed and its corresponding latitude (JLI) averaged over the first 10
years (1980s, black cross) and the last 10 years (2010s, red cross) are also shown. In summer and
autumn, the maximum zonal wind weakens, and the center of the jet stream moves northward
(0.2° in JJA and 1.0° in SON). A weakening of the maximum zonal wind speed is also found in
winter. Spring is unique as its maximum zonal wind is amplifying. The center of the jet stream
moves southward in winter and spring.
Figure 2 gives another overview of the NPWJ and its variation from both horizontal and vertical
perspectives by showing the 200 hPa zonal wind (altitude of maximum wind speeds) and the
latitude-height distribution of the zonal wind averaged over the studied longitudinal segment of
north Pacific in JJA. Here, the labeled contour lines and numbers indicate the zonal wind
magnitudes and the shaded color values indicate the decadal trend of the zonal wind during
1980-2019. Clearly, the Pacific jet stream (30-45N) has been weakening over the period shown
by all reanalysis data. Furthermore, the weakened zonal wind center is slightly southward
relative to the maximum zonal wind center, thereby pushing the jet stream center northward.
Similar plots for SON are shown in Fig. S1, which shows that NPWJ has weakened more
profoundly, and its center has moved more northward compared to the case in JJA, which is
consistent with the feature revealed in Fig. 1.

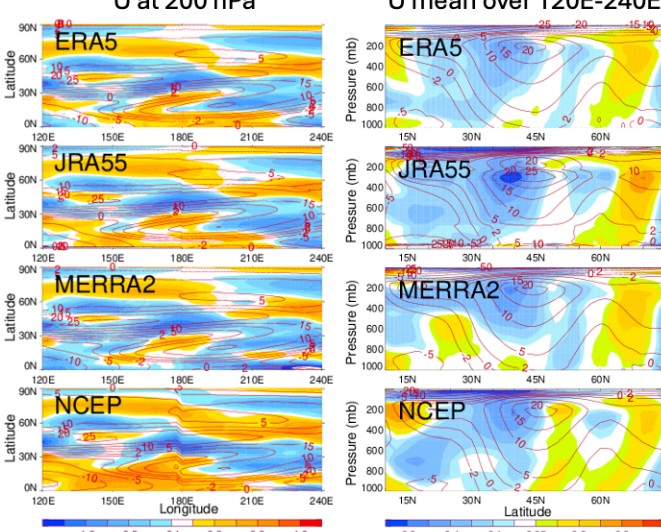

**Figure 2. NPWJ strength and location shown by zonal wind U (m s$^{-1}$, lines) and its decadal trend (m s$^{-1}$**
**decade$^{-1}$, shaded) in June-July-August (JJA, summer) during 1980–2019 from the four reanalysis datasets at**
**200 hPa (left column) and latitude-height distribution (right column).**
The time series of 200 hPa seasonal and NPA averaged zonal wind for the period 1980- 2019 is
shown in Fig. S2 for the four reanalysis datasets. It is noteworthy that, although the maximum
zonal winds in the North Pacific westerlies vary seasonally, the significant zonal wind trends




generally occur within the range of 30–45 N throughout the year, as examples shown in Figs. 2
and 3 for JJA and in Figs. S1 and S3 for SON. We measure the significance of any detected
trends in terms of *p*-values calculated using Kendall's (tau) rank correlation, which is a
nonparametric method, i.e., it makes no assumptions about the underlying distribution of the data
and its rank-based measures are not affected by extreme values. The smaller the p-value, the less
likely the trend found in the data is from random fluctuations alone. All four reanalysis datasets
show a clear weakening trend of NPWJ in summer and autumn with *p*-values in most cases less
than 0.1 (a criteria used in Dong et al. 2022), respectively, whereas no significant trends are
apparent in spring and winter.
**3.2 Contribution of atmospheric forcings to the trend of the NPWJ using CMIP6 results**
In this section, we use the CMIP6 model experiments to answer the 3rd science question raised in
section 1 by examining how the NPWJ responds to various atmospheric forcings. We present the
detailed responses of the NPWJ to various atmospheric forcings in JJA in the main text and in
SON in the supplementary material.

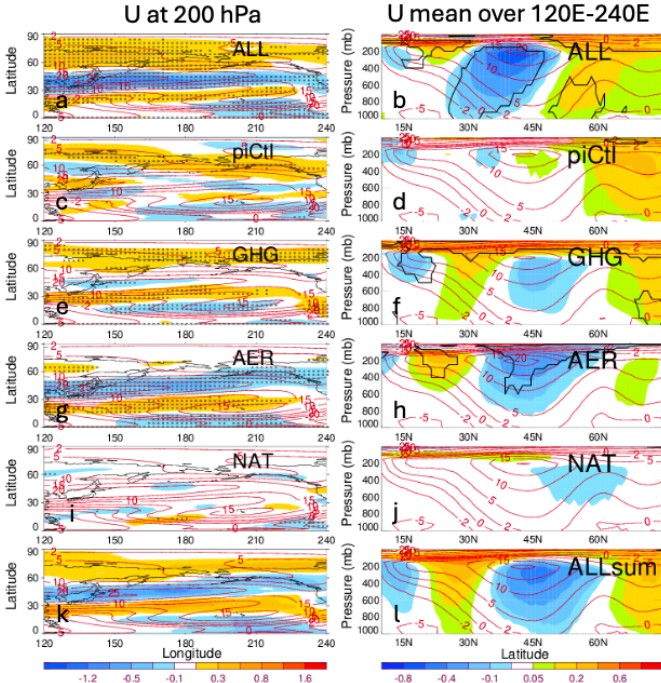

**Figure 3. Similar to Figure 2 but using eight CMIP6 model results. The data shown here are multi-model
ensemble mean in JJA during the period of 1980-2014. The NPWJ strength and location and its decadal trend
are shown not only with all atmospheric forcings (ALL), but also with individual single forcing of GHG,
AER, and NAT. Also shown here are the NPWJ information associated with pre-industrial control run
(piCtr) and the lump sum of the three single forcings of GHG, AER, and NAT (ALLsum). Crosses in a, c, e, g,
and i and black lines in b, d, f, h, and j indicate regions where trends are statistically significant at the 10%
level using the Mann–Kendall test.**





We first performed a similar analysis as shown in Fig. 2 but using the multi-model ensemble
mean (MMM) from the eight CMIP6 models (see Section 2) for the period 1980-2014. The
results presented in Fig. 3 provide an in-depth analysis of the NPWJ response to various
atmospheric forcings in addition to all-forcings. Using results of the CMIP6 sensitivity
experiments (described in section 2), we can answer the following science questions: How much
have the changes in anthropogenic aerosols, greenhouse gases, and natural changes exerted an
influence on the NPWJ strength and location over the period of 1980-2014 on a seasonal basis?
Which forcing is the driving forcing and why? What can the changes in atmospheric temperature
and radiation fields tell us the NPWJ trends?
ALL, GHG, AER and NAT in JJA is shown in Fig. 3 to highlight their similarities and
discrepancies. Given the reanalysis data are representative of observational characteristics, the
features of the NPWJ shown in Section 3.1 can be used to evaluate the CMIP6 MMM simulation
with all forcings included. Overall, the CMIP6 MMM ALL results support the conclusion of a
weakening trend of NPWJ revealed by the reanalysis data, although the trends of the zonal-wind
are more stratified with latitudes with the decreasing trend concentrated in the mid-latitudes of
30-45N for all longitudes in the model results. The responses of the NPWJ to these potentially
important forcings indicate that aerosols are the primary driver of the decadal trends because the
magnitude is greatest in the AER plot (Fig. 3g–h). The same plots for the SON season are shown
in Fig. S3.
To interpret the forcing signal with high confidence, we need to remove the residual climate drift
(or natural climate internal variability) in the experiment since the individual climate signals
produced by the proposed perturbations could be small compared to the internal climate
variability. The internal variability means the system internal evolution without external forcing.
The climate impacts of atmospheric forcings can then be diagnosed by subtracting the perturbed
runs from the historical climate and evaluated against internal variability diagnosed from piCtl.
The eight selected CMIP6 models not only performed at least 3 ensemble simulations, but also
ran pre-industrial control simulations for at least 150 years. Figures 3c-d show the multi-model
average piCtl for JJA over the final 35 years (and Fig. S3c-d for SON), which shows no
statistically significant trends of the NPWJ. Thus, the apparent decreasing trend in NPWJ in
summer and autumn is caused by forcings other than by model internal variability.
DAMIP designed the GHG, AER, and NAT experiments using the "only" approach, i.e., only the
forcing of interest was varied for simulation, while all other forcings were held constant at pre-
industrial values (Gillett et al., 2016). The validity of the additivity assumption has been
considered in studies using DAMIP simulations (Gillett et al., 2016), that is, the climate response
to all forcings is equal to the sum of the responses to the individual forcings. However, the clear
differences between Fig. 3a (response to all forcings) and Fig. 3k (sum of the responses to the
individual forcings in Figs. 3e, g, and i), and between Figs. 3b and 3l, suggest that in addition to
GHG, AER, and NAT, some other forcings also contribute to climate change in the historical
(i.e., all) experiments. One potential forcing could be ozone, since time-varying ozone
concentrations were used in the historical experiment (Eying et al., 2016), while the pre-
industrial stratospheric and tropospheric ozone climatology was used in the radiative scheme of
the DAMIP GHG experiment (Gillett et al., 2016). Here, the trends produced by historical all
forcings are the most pronounced, with those of the NPWJ being more consistent with those of





the reanalysis data in terms of jet strength, core location, and vertical shape. This emphasizes the
importance of accounting for the impact of time-varying ozone radiative forcing on the distant
ocean-atmosphere dynamical fields in the simulations.
Figure 4 shows the decadal trends for the summer NPWJ for the four reanalysis datasets and the
eight CMIP6 models under different atmospheric forcings, where the contribution of each
individual model is presented together with the CMIP6 MMM results. Each symbol represents an
individual reanalysis or CMIP6 model dataset. For each CMIP6 model its own ensemble mean is
shown (see Table 2 for the number of members). The crosses for ALL, GHG, AER, and NAT are
the corresponding CMIP6 MMM results. The four reanalysis datasets and the eight CMIP6
MMMs all show a clear decreasing trend, up to -0.45 m s$^{-1}$ decade$^{-1}$ in JJA. Further analysis
tracing back to the single forcing perturbations in CMIP6 shows that only the single forcing of
AER causes a decreasing trend in U wind, but its amplitude is much weaker than the values of
overall forcings. The large difference between the sum of the three individual forcing simulations
and the historical control simulation supports the potential nonlinear interaction between the
atmospheric forcings shown in Fig. 3. A similar analysis in SON (Fig. S4) shows a larger
decreasing trend of zonal wind in the reanalysis data (up to -0.68 m s$^{-1}$ decade$^{-1}$), but this is not
the case in the CMIP6 results.

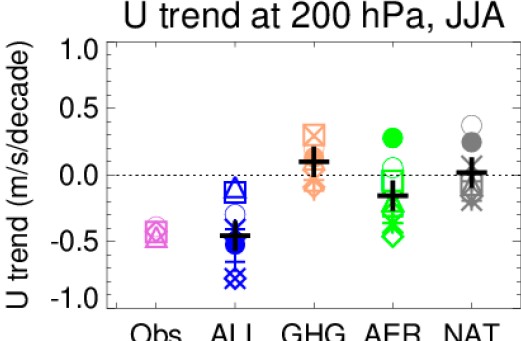

**Figure 4. Decadal zonal wind trends (m s$^{-1}$ decade$^{-1}$) at 200 hPa in JJA over the North Pacific using**
**four_reanalysis data (Obs) and eight CMIP6 model ensemble means, calculated by different forcing**
**simulations (ALL, GHG, AER, NAT). Symbols for the reanalysis data: ERA5 (open circle), JRA55**
**(Diamond), MERRA2 (Triangle), and NCEP (Square). Symbols for the model data: BCC-CSM2-NR (open**
**circle), CanESM5 (diamond), CNRM-CM6-1 (triangle), GISS-E2-1-G (square), HadGEM3-GC31-LL (cross**
**sign), IPSL-CM6A-LR (plus sign), MIROC6 (asterisk), and MRI-ESM2-0 (filled circle). The multi-model**
**mean (MMM) of CMIP6 model ensemble results is shown by thick black plus sign.**
What are the physical mechanisms behind the NPWJ trends? We first examine the responses of
the meridional temperature gradient at 500 hPa (Fig. 5a–d) to all forcings and to the three single
forcing experiments in JJA. The level of 500 hPa is a proxy for a layer from the surface to 200
hPa as the reduction of the meridional temperature gradient (MTG) in the lower troposphere
leads to a reduction in the vertical shear of the U-wind, and thereby weakening the upper
tropospheric jet through the balance of thermal winds. These plots show that the significant
weakening trend in NPWJ shown by the overall gradient is mainly due to aerosol forcing. The
dominant role of aerosol forcing is also shown in the latitude-height distribution of the MTG
(Fig. 5e–h). The MTG has been demonstrated to be the fundamental physical mechanism driving





the motion of the westerly jet (Francis et al., 2012; 2015; Liu et al., 2012; and Overland et al.,
2010). The larger the MTG is, the stronger the westerly jet stream becomes. Over the period of
1980-2014, evidence suggests the slowing jet stream coincides with Arctic warming (Francis and
Vavrus et al., 2015). The following equation describes the connection between vertical thermal
wind shear and the horizontal MTG (Holton 1992; Rotstayn et al., 2014).

$$\frac{\partial u}{\partial p} = \frac{R}{fp}\left(\frac{\partial T}{\partial y}\right)_p$$

Here $u$ is zonal-mean zonal wind (m/s), $p$ is pressure (Pa), $R$ is the gas constant for dry air, $f$ is
the Coriolis parameter, and $(\partial T/\partial y)_p$ is zonal-mean MTG on a constant pressure surface. In order
to calculate the thermal winds at each layer, it is necessary to integrate the calculated results
upward from the surface. In other words, the thermal wind at 200 hPa is caused by the thermal
energy changes in the atmosphere for $p > 200$ hPa. The results are shown in Fig. 5i-l, indicating
that the change in atmospheric aerosols is the main driver affecting the horizontal temperature
distribution, thereby affecting the upward thermal winds. Similar results in SON are given in Fig.
S5.

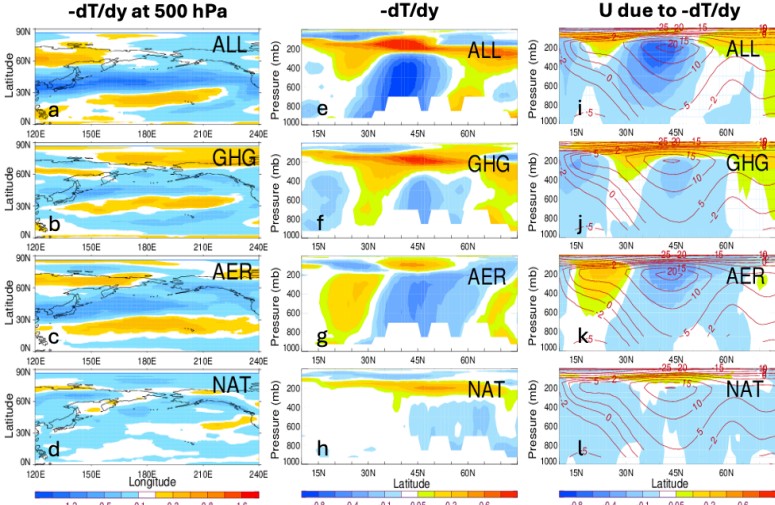

**Figure 5. The summer (JJA) decadal trend of MMM over 1980–2014 from CMIP6 CMIP and DAMIP simulations for (a-d) -dT/dy (a.k.a. meridional temperature gradient, MTG) at 500 hPa (K per 1000 km decade⁻¹), (e-h) latitude-height zonally averaged -dT/dy over north Pacific sector (120°E - 240°E), and (i-l) zonal winds derived based on the thermal wind balance from the cross-section of MTG in middle column. The contour lines on (i-l) are the corresponding MMM zonal wind.**

Why is aerosol the driving forcing? The spatial distribution of aerosol trends during 1980–2014
is not uniform, leading to differences in regional SSR changes, which in turn cause changes in
horizontal temperature gradients (Allen et al., 2020; Dong et al., 2021; Rotstayn et al., 2013),
which in turn affect the atmospheric circulation, including the strength and location of the jet
stream (Allen and Ajoku 2016; Chen et al., 2018; Rotstayn et al, 2014; Shen et al., 2018; Undorf
et al., 2018). Figure 6 shows the MMM results of the JJA AOD and its trend over the Northern
Hemisphere (a), the timeseries of AOD over East Asia (75°–130°E and 20°–45°N) (b), MMM
SSR trend (c,f,i,l), MMM T trend at 500 hPa (T500hPa, d,g,j,m), and MMM MTG (dT/dy,



e,h,k,n) driven by all forcings and by the three single forcings. A meridional gradient index of
AOD and SSR is examined as the area-averaged difference between two regions located to the
south and north of the upstream of Pacific climatological jet core, i.e., 20°–45°N (South and East
Asia, SEA) and 45°–60°N (North Asia, NOA) over 75°–130°E. The region for T500hPa analysis
is moved further northeast in the North Pacific region (i.e., 30°–45°N for the south box and 45°–
60°N for the north box over 120°–240°E) to consider the downstream influence of aerosol
emissions. We can clearly observe a solar dimming occurred in SEA (i.e., decreasing SSR trends
in c and i), which is due to a large increasing trend of Asian AOD over China and India,
accompanied by a solar brightening (i.e., increasing SSR trends in c and i) occurring in NOA,
which can be traced back to a significant decrease in AOD over Europe (a). This creates a
spatially inhomogeneous downwind temperature distribution at T500hPa in the south and north
boxes (d and j), which results in a weakening of the MTG over the NPWJ region (e and k),
which consequently leads to a weakening of the vertical thermal wind. A similar aerosol-
temperature-MTG-thermal wind relationship during SON is shown in Fig. S6. On the other hand,
the response of climate system to the increase in greenhouse gases (GHG) faces a potential "tug-
of-war" feature. The influence of the reduced lower-tropospheric MTG due to amplified Arctic
warming associated with GHG (Coumou et al., 2015, 2017, 2018) may counter an increased
MTG due to enhanced convection and latent heat release in the upper troposphere (IPCC 2013;
Shaw and Voigt, 2015; Boucher et al., 2013).

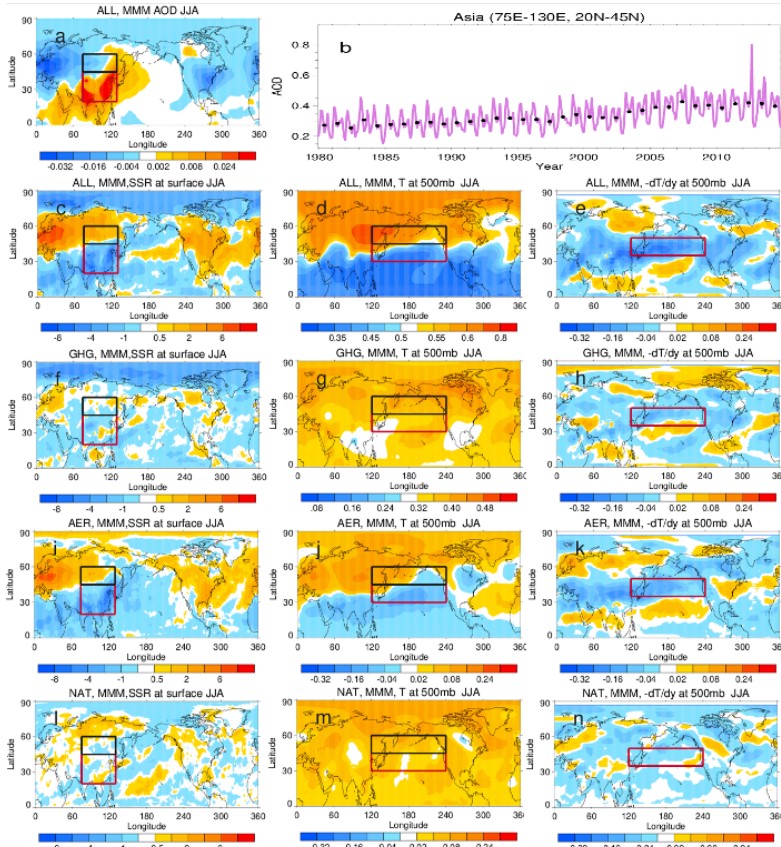




**Figure 6. Spatial patterns of aerosol, radiation, temperature, and meridian temperature gradient in summer (JJA) using CMIP6 MMM data during the period of 1980-2014 to illustrate the connection among these fields. (a) AOD trend spatial distribution, (b) timeseries of AOD (monthly mean in purple and annual mean in black) averaged over the Asia area within the southern box in (a), (c, f, i, j) downward surface shortwave radiation trend (SSR, Wm$^{-2}$ decade$^{-1}$), (d, g, j, m) temperature trend at 500 hPa (T500hPa, K decade$^{-1}$), and (e, h, k, n) MTG at 500 hPa (K per 1000 km decade$^{-1}$) from All, GHG, AER, and NAT, respectively. The red and black boxes in SSR and T500hPa highlight the regions used to define their gradient indices (south box minus north box). The red boxes in the MTG highlight the areas with eventual perturbations to the NPWJ due to aerosol changes.**

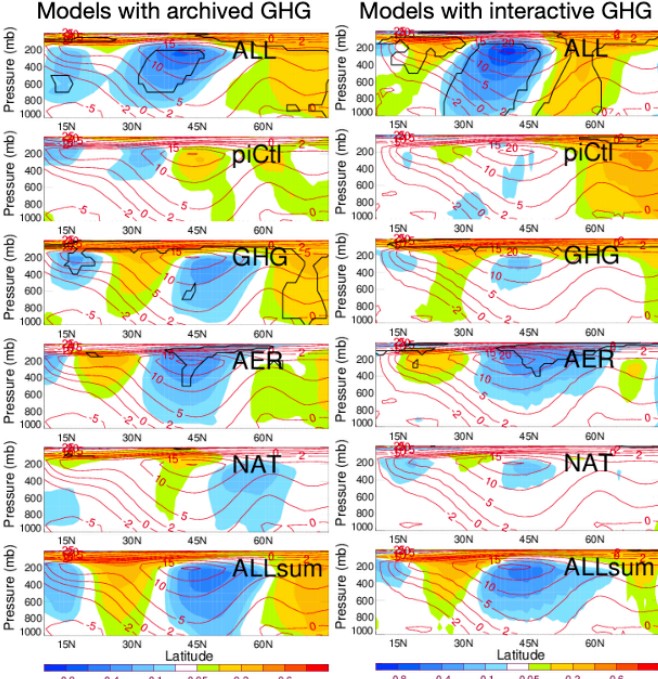

**Figure 7. Similar to Fig. 3 but dividing the eight CMIP6 models into two groups without (left column) and with (right column) interacting chemistry.**

The datasets of time varying GHGs (i.e., $CO_2$, $CH_4$, $N_2O$, $O_3$, and ozone-depleting substances) recommended by CMIP6 were applied by four CMIP6 DAMIP models (i.e., BCC-CSM2-MR, CNRM-CM6-1, MIROC6, and HadGEM3-GC31-LL). These CMIP6 recommended data were provided on a latitudinal and seasonal basis and were based on AGAGE and NOAA observation networks, firn and ice core data, archived air data, and a large set of published studies. The other four CMIP6 DAMIP models accounted for GHGs differently – CanESM5 (similar archived CMIP6 recommended GHGs except interactive $CO_2$), GISS-E2-1-Gits (GHGs obtained from its own previous interactive chemistry), MRI-ESM2-0 and IPSL-CM6A-LR (GHGs calculated from interactive chemistry). To further quantify GHG contribution using archived GHG data or interactive chemistry, we analyze the CMIP6 results separately in two model groups (GHGs from CMIP6 datasets vs from CMIP6 model interactive chemistry). Both sets of CMIP6 models support the conclusion that the NPWJ has been weakening during the summer period over the





past three decades with AER being the strongest driver (Fig. 7). The figures also reveal the time-
varying ozone radiative forcing that affects the NPWJ, regardless of the GHG used, as shown by
the difference between ALL and ALLsum. Although the above conclusions are highly robust, the
two groups of CMIP6 models do show different NPWJ trends at certain levels, particularly in the
lower troposphere and under perturbations of internal variability and GHG single forcing. The
performance of the two groups of CMIP6 models differs more significantly in autumn, as shown
in Fig. S7, particularly for single forcing, although both groups of models still show a weakening
trend in the NPWJ in all forcing simulations. Considering that the NPWJ trends simulated by the
CMIP6 models when using interactive chemistry are more similar to those of the reanalysis, such
performance suggests that interactive chemistry plays an important role in simulating long-term
feedbacks between atmospheric composition changes and atmospheric dynamics.
**4. Discussion**
An atmospheric reanalysis assimilates historical atmospheric observational data spanning an
extended period using a single consistent analysis scheme throughout. Although errors can be
created by degradation, replacement, or modification of instruments (e.g., satellites), as well as
by changes in methods of observation (Trenberth et al., 2001), reanalyses can often be thought of
as the best estimate available for many atmospheric variables, particularly the winds (Kaiser-
Weiss et al., 2015) and temperature of the atmosphere as they are among the reanalysis data that
are best constrained by observations. To highlight the significance of zonal wind trends, a rank-
based nonparametric method Mann–Kendall statistical test (Mann 1945; Kendall 1975) has been
applied. Unlike the fields of winds and temperature, only ERA5 and MERRA2 reanalysis
includes the available aerosol observational data. There are clear advantages of using aerosol
reanalysis, which ensures episodic events (e.g., fires, dust storms, volcanoes) are frequently
updated (e.g., hourly rather than monthly) and satellite observations of both meteorological and
aerosol data are incorporated into the aerosol reanalysis through data assimilation (Xian et al.,
434  2022).
All eight CMIP6 models mentioned in section 3 have a long history in scientific studies. The
similarity in NPWJ strength, location, and trend seen in the CMIP6 multi-model simulations and
the structure seen in the four reanalysis products suggests that common factors may be
responsible. Regional variation in anthropogenic aerosol emissions (i.e., a large increase in Asia
and a large decrease in Europe) inevitably induces the changes in NPWJ, which has been shown
in Figs 6 and 5. The conclusion that anthropogenic aerosol precursor emissions were the primary
driver of the weakening of the summer to autumn NPWJ over the period of 1980-2014 is robust
by reviewing the distinctive fingerprint of AER forcing identified in CMIP6 DAMIP experiment
in the context of the errors in the models. Specifically, this conclusion is drawn when the
interactive chemistry of GHGs is taken into account and the potential internal variability of the
models is small compared to that of AER forcing.
**5.  Conclusion**
This study focuses on the impact of long-term trends in atmospheric composition and their
connection to the strength, location, and trend of the Pacific westerly jet using various reanalysis
data, which assimilate current and past observational datasets including various satellite
products, where causal relationships are elucidated using CMIP6 model simulations that couple
atmospheric and oceanic systems. By examining the jet stream latitude index, a PDF distribution



of the maximum westerly wind speed and the corresponding latitude determined based on the
daily mean zonal winds in the North Pacific from 1980 to 2019 studied by four reanalysis data,
we found that the NPWJ has a distinct seasonal pattern. The jet is strongest and most southerly in
winter, while it is weakest and most northerly in summer, with spring and autumn falling in
between. The four-reanalysis data also reveal with high confidence a weakening of the strength
and a northward shift of the NPWJ in summer and autumn during the studied period.
Similar analyses using eight CMIP6 models all support the observed weakening trend of the
NPWJ, indicating that the CMIP6 models are well-equipped for climate research and the
conclusions drawn are robust. Further studies using CMIP6 results from simulations driven by
various important atmospheric forcings (such as GHG, AER, and NAT) show that anthropogenic
aerosol forcing is the primary forcing for the decadal changes in mid-latitude atmospheric
circulation in recent decades. The linkage of aerosol forcing and NPWJ changes is mainly
manifested in the change of the meridional temperature gradient, which is the physical
mechanism for the formation of thermal winds. During 1980-2014, the inhomogeneity of
anthropogenic aerosol changes (increase in Asia, decrease in Europe) has led to a dipole pattern
in the trends of AOD, SSR, and T, that is, an increase in the south-north gradient of AOD and a
decrease in SSR over Asia, which has led to a decrease in MTG downwind of the North Pacific
where the NPWJ exists.
The CMIP6 models used capture observation-based reanalysis NPWJ changes better when
considering all atmospheric forcings than when the responses of individual forcings are
combined in DAMIP GHG, AER and NAT simulations, suggesting the importance of
considering time-varying ozone radiative forcing in long-term Earth system climate studies.
Introducing interactive chemistry into the models is also necessary for simulating decadal-scale
atmospheric dynamics. The performance of the two groups of CMIP6 models with and without
interactive chemistry differs particularly in autumn, explaining why the NPWJ characteristics
shown by the CMIP6 results are more consistent with reanalysis data in summer than in autumn.
From both scientific and policy perspectives, it is necessary to assess how anthropogenic
aerosols affect atmospheric circulation in the coupled atmosphere-ocean system on seasonal to
multidecadal scales, as well as the impact of atmospheric feedbacks on weather and long-range
transport of aerosols. In addition, understanding the consequences of past human activities on the
environment and climate is essential for making high-confidence predictions of potential future
impacts. Rapid economic growth in Asia over the past few decades has led to a significant
increase in air pollution across Asia and a weakening of the North Pacific jet stream, which
brings profound impacts on a range of atmospheric phenomena and the long-range transport of
air pollution to North America and beyond through pollution-weather-climate interactions, which
will be explored in an accompanying paper.
**Acknowledgements:**
The authors thank the four institutional centers (ERA5, JRA55, MERRA2, and NCEP) for
providing long-term reanalysis data, as well as the eight modeling groups (BCC-CSM2-MR,
CanESM5, CNRM-CM6-1, GISS-E2-1-G, HadGEM3-GC31-LL, IPSL-CM6A-LR, MIROC6,
and MRI-ESM2-0) who contributed to the CMIP6 multi-model intercomparison. This research
was funded by the NASA Atmospheric Composition Modeling and Analysis Program (ACMAP



80NSSC23K1000). Computational resources supporting this research were provided by the
NASA GMAO SI Team and the High-End Computing (HEC) Program at the NASA Center for
Climate Simulation (NCCS) at Goddard Space Flight Center (GSFC).
**Data Availability:**
The four reanalysis datasets and eight CMIP6 simulation datasets used in this study are publicly
available. Detailed data acquisition information for the four reanalysis datasets is listed in Table
1. The eight CMIP6 simulation datasets were downloaded from the Centre for Environmental
Data Analysis (CEDA) at https://help.ceda.ac.uk/article/4801-cmip6-data.
**Author Contributions:**
H.B. took an overall responsible for the experiment design, data collection, and analysis. All
authors contributed to the writing of the manuscript.

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
