# Peer review of "Investigating recent decadal trends in the Pacific westerly jet in response to various atmospheric forcings using CMIP6 model results and reanalysis data"

_EGUsphere, 2025_

## Author Comment (AC1)

The reviewers raised a very challenging question: how to differentiate our research from that of Kang et al. (2024) and clarify the novelty of our findings. Kang et al. (2024) also used the DAMIP model results and concluded that aerosol forcing played a dominant role in the weakening of the NPWJ during the summer period 1980-2014. Unfortunately, this is precisely the main finding of our study.

We believe our research is worthy of publication because it uses a different methodology than Kang et al. (2024) to confirm the finding. Furthermore, our study finds that the NPWJ changes obtained in CMIP6 historical simulations, which consider all atmospheric forcings, differ from the combined response of individual forcings in the DAMIP hist-GHG, hist-aer, and hist-nat simulations. This finding of non-additive effect is novel and important in itself, as it challenges the validity of the additivity assumption (i.e., that the climate response to combined forcings equals the sum of the responses to individual forcings) used in DAMIP simulation studies (Gillett et al., 2016). We hypothesize that the most likely reason for this non-additivity is the lack of consideration of time-varying ozone radiative forcing in the DAMIP single-forcing experiments. In the DAMIP design, the single-forcing experiment hist-GHG only considers the contribution of time-varying well-mixed greenhouse gases, while keeping tropospheric and stratospheric ozone concentrations constant. On the other hand, the CMIP6 historical experiment uses time-varying atmospheric ozone. Our study suggests that more work is needed to test whether ozone forcing is indeed relevant. If the editors and reviewers agree with this argument, we will revise the manuscript to explain our innovative work with the above discussion and address the reviewers' other comments.

If publication is deemed necessary for further in-depth research and validation of our hypothesis (i.e., that ozone forcing is relevant), we intend to use the hist-stratO3 and hist-totalO3 experimental data from the LESFMIP project (Smith et al., 2022) (an extension of the DAMIP project) as suggested by Reviewer 1. Reviewer 1 commented that without compelling evidence that ozone is a significant influencing factor, the possibility of non-additive effects of various DAMIP single-forcing contributions cannot be ruled out. Unfortunately, we are currently unable to definitively determine the driving causes based on the data provided by the CMIP6 DAMIP project. To clearly reveal the underlying causes, we recently accessed the CMIP ESGF online archive (https://aims2.llnl.gov/search) suggested by Reviewer 1 and found that the new LESFMIP data has not yet been submitted. We also checked other archive websites where CMIP6 data is available, such as https://help.ceda.ac.uk/article/4801-cmip6-data and https://esgf-ui.ceda.ac.uk/search. On the latter website, we selected CMIP6Plus in the "Select a project menu" and then found the LESFMIP project under the general tab under "filter with facets". The good news is that some data has already been submitted under this project. The bad news is that currently only one model (HadGEM3-GC31) has submitted data (see the "Source ID" list). Furthermore, this model has only submitted data for hist-lu, hist-piAer, and hist-piVolc (see the "Experiment ID" list), and has not yet submitted data for hist-stratO3 and hist-totalO3. Perhaps we need to be a little more patient. If the editor and reviewers suggest that further research on the relevance of ozone is necessary, we request an extension of the revision deadline to allow time to wait for the release of the LESFMIP data.

Please tell us which of the above methods you recommend, or whether you believe that, given the article published by Kang et al. (2024), we need to make further efforts to highlight the novelty of our research beyond what we have presented here.